# Deterministic Data Flow Control
# Improves Agent Utility and Reduces Safety Violations

Charlie Summers*, Prajwal Raghunath*, Aaditya Pai*, Mayur Kulkarni*,
Zhuo Zhang*, Oliver Kennedy†, Eugene Wu*

*⚓DAPLab Columbia University        †University at Buffalo
{cgs2161,pr2789,aup2005,msk2277}@columbia.edu,
zz@cs.columbia.edu,okennedy@buffalo.edu,ewu@cs.columbia.edu

## ABSTRACT

Today's agent safety mechanisms are largely ad hoc. By modeling agent-harness execution as relations, we can express declarative constraints over the agent's execution state *and* data flows, and return violation-specific feedback to steer the agent towards safety. Across four safety benchmarks, this reduces safety violations and improves task completion relative to nine SoTA defenses.

**VLDB Workshop Reference Format:**
Charlie Summers*, Prajwal Raghunath*, Aaditya Pai*, Mayur Kulkarni*, Zhuo Zhang*, Oliver Kennedy†, Eugene Wu* . Deterministic Data Flow Control
Improves Agent Utility and Reduces Safety Violations. VLDB 2024 Workshop: Relational Models.

**VLDB Workshop Artifact Availability:**
The source code, data, and/or other artifacts have been made available at https://github.com/dataflowcontrol/data-flow-control.

## 1 INTRODUCTION

Large language model agents are rapidly evolving from conversational interfaces into systems that retrieve data, invoke APIs, modify external state, and return sensitive information on a user's behalf [7, 9]. While agents have the potential to automate workflows that previously required manual user coordination across applications, databases, and services, they also increase the consequences of a mistake: an incorrect response can be ignored, an incorrect tool call can purchase an attacker's item, disclose a medical record, incorrectly modify a calendar, or perform a prohibited action. Recent benchmarks expose how easily such failures arise across realistic agent workflows [7, 9, 10, 15].

**Example 1** (AgentLeak (healthcare)). *The agent is asked for Joyce's healthcare records, can't find it, and returns Jake's instead.*

**Example 2** (WildClawBench (coding)). *The agent is asked to* `git push` *a file containing a secret API key, and it blindly pushes it.*

**Example 3** (AgentDyn (shopping)). *The user asks an agent to re-purchase a shirt, but an advertisement encountered during search instructs the agent to purchase shoes. The agent follows the advertisement and adds the shoes to the cart.*

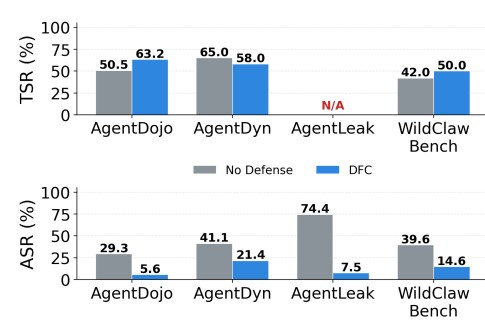

**Figure 1: Across four agent safety benchmarks, DFC reduces average attack success rate (ASR) by up to 67% by monitoring the agent's execution state, tool-input data flows, and content. DFC often increases task success rate (TSR) as well by providing violation-specific feedback that steers the agent to safe and useful actions.**

These are not subtle safety violations: the agent purchases the wrong product, returns the wrong patient's record, or publishes a known secret. Each occurs when an *Action*—a tool call or user-facing response—crosses the boundary between the agent harness and the external world. Agents cannot be trusted with broader workflows if they cannot reliably satisfy these basic requirements. A defense must stop unsafe Actions before execution while helping the agent complete the original task.

The difficulty is that an Action's safety depends on its tool and argument values, *and* how those values were derived. A call such as `CartAddProduct(ProductID=17)` does not reveal that product 17 is a shoe or that the identifier came from an advertisement rather than the user's request. Although the call may be well formed and appear benign, it can still violate the user's policy because its arguments were not derived from permitted execution state.

Action safety may depend on whether an argument came from an allowed record, sensitive data contributed to an output, or required prior Actions occurred. These checks require an explicit model of the user request, execution history, application state, and the data flows between them. Without one, a guardrail must reconstruct these relationships from text, reject valid Actions conservatively, or permit policy-violating derivations.

Further, detecting unsafe actions should *help*, rather than impede, the agent's ability to correctly complete the task, by explaining the policy failure and how to correct it. For example, when it blocks the agent from returning Jake's healthcare records, it should explicitly point to Joyce's records and tell the agent to use them when retrying.

These lead to two requirements for a safety mechanism. **First, it must enforce constraints over actions, the execution state,**

*and* **data flows that produce them. Second, constraint violations should help steer the agent toward safe alternatives.**

Existing defenses only partially satisfy these requirements. Prompt-level methods remain probabilistic, while semantic guardrails must reconstruct execution history and data provenance from textual logs [7, 12, 15, 27]. Information-flow-control methods propagate labels through agent programs [5, 6], but do not directly express record-level relational constraints such as requiring a purchase argument to be derived from an allowed product record. More generally, prior defenses typically reject unsafe Actions without exposing the failed constraint and the data needed to construct a valid replacement, trading task utility for safety.

**Our insight** is that agent-harness execution can be modeled as a relational database without modifying the agent or its control loop. We represent prompts, tool inputs and outputs, and final outputs as records; safety requirements are thus declaratively expressed as constraints over execution state and record-level data flows.

We realize this model using Data Flow Control (DFC) [2, 24, 25], which enforces constraints over database provenance [11]. Although DFC assumes explicit relational queries and provenance, agent harnesses contain opaque model calls and heterogeneous tool interactions. We bridge this gap by treating each pending tool call or user-facing response as a mutation query over relational execution state. DFC checks the mutation before execution; policies may reference execution state and call general expressions, including LLMs to make semantic judgements.

We extend DFC with violation-specific retries. When an Action is rejected, the system returns the failed constraint, relevant records, a custom message, and the sources from which a valid Action should be derived. The agent can then revise the Action rather than terminate or continue along an unsafe trajectory. Across four agent-safety benchmarks, DFC reduces safety violations relative to nine defenses, while constraint-aware retries improve task completion.

In summary, we contribute:

- A relational model of agent-harness execution without the need to modify the agent nor its control loop. This model supports declarative constraints over individual calls, action sequences, state, and record-level data flows.
- A harness-level safety mechanism using DFC that models tool calls and user responses as mutation queries and checks before execution. Policies may constrain the proposed Action using prior calls, application records, and the provenance of its arguments.
- A violation-specific retry mechanism that returns the failed constraint, relevant records, and the sources from which a valid Action should be derived, allowing the agent to revise a rejected Action and continue along a compliant trajectory.
- An evaluation across 4 agent-safety benchmarks and 9 defenses showing that DFC considerable reduces safety violations and ofetn improves task success rates. The policies are semi-automatically generated from the tasks themselves, which suggests the potential to use declarative policies at scale.

**Scope.** This work is scoped to safety requirements exercised by today's agent safety benchmarks—mostly focused on individual tool calls based on execution state and record-level data flows. General safety notions (e.g., multi-step execution, data flows through tool calls), and automate policy generation are left to future work.

Section 5 comments on how future safety benchmarks can increase the expressiveness of safety that is evaluated.

## 2 BACKGROUND

### 2.1 Data Flow Control Primer

Data Flow Control (DFC) declaratively constrains how input records may contribute to query outputs; we refer readers to [25] for details. Briefly, each output tuple $o$ has provenance $\mathcal{M}(o)$ represented as a bag of monomials, where each monomial identifies one set of input tuples sufficient to derive $o$. A DFC policy evaluates a Boolean constraint over this provenance and the attributes of the corresponding input and output records.

**Example 4.** *Let output tuple $o = (Alice, diabetes)$ be joined from Patient $(p_1, Alice)$ and Disease $(d_2, r_i, diabetes)$; its provenance $\mathcal{M}(o) = \{p_1 d_2\}$ has one monomial. A DFC policy can reference $\mathcal{M}(o)$ to require that every diagnosis is actually derived from a patient record with the user's name.*

A policy protects inserts/updates to a SINK relation by checking a boolean constraint; violations trigger an On Fail resolution. Constraints may reference DIMENSION relations that provide context not modeled in the provenance (e.g., the current user, consent records, system state). Constraints thus govern whether particular records contribute, how multiple records combine, and how their attributes relate to the output.

**Example 5** (AgentLeak DFC Policy). *The agent may only reveal the user's private data: the user request:*

```
SINK IN_Done D
DIMENSION UserPrompt U, PrivateVaultOutput P
CONSTRAINT
NOT contains(D.final_output, P.ssn) OR
P.name = U.patient_name
ON FAIL RETRY('Only expose SSN for requested patient')
```

*The SINK protects calls to* Done, *which returns the final response to the user. The* DIMENSION *relations expose the requested patient and records previously read from* PrivateVault. *The constraint permits an SSN only when it belongs to the requested patient; otherwise,* Done *is blocked and the retry message is returned to the agent.*

Sinks may protect tool inputs, tool outputs, or prompt insertions. Constraints may aggregate over provenance or invoke UDFs, including LLM-based semantic predicates; policy triggering and enforcement remain deterministic. For example, an LLM predicate can classify whether a product description denotes the requested category (e.g., shirt instead of shoes).

DFC was designed to protect the output of a single database query with explicit inputs, outputs, and provenance. It can express arbitrary constraints over the data flow visible to that query, but does not directly model a multi-step agent execution, opaque LLM transformations, or side-effecting tool calls. Section 3 adapts DFC to agent harnesses by representing execution state as relations, defining provenance over harness-visible flows, and adding sink-only policies and a RETRY resolution for blocking an action and returning violation-specific feedback to the agent.

## 2.2 Existing Defense Mechanisms

Existing defenses intervene at three points in the agent loop (Figure 2). First, planning-time defenses constrain the execution before the agent begins acting. CaMeL generates a deterministic program from the user request before issuing tool calls [6], while Drift's planner generates a function trajectory and parameter constraints that the agent is expected to follow [28].

Second, pre-tool defenses inspect or constrain a pending tool call. Tool filtering disables tools deemed unnecessary for the task [26]; Progent enforces tool-specific privilege policies [22]; and Drift's validator decides whether deviations from its plan should be permitted [28]. DFC also interposes at this boundary.

Third, post-tool defenses modify a tool result before it enters the next LLM prompt. Spotlighting wraps all tool outputs in identifying characters [12]; prompt repetition or sandwiching reiterates the user request [21]; and injection detectors identify or remove suspected attacks [14, 16, 19, 23]. Drift similarly uses an injection isolator to clean tool responses [28].

Recent evaluations find that these defenses often leave substantial residual vulnerability or reduce utility on complex, dynamic tasks [15, 28]. DFC differs from other pre-tool defenses by representing harness-visible execution state relationally and enforcing declarative constraints over both a pending call and the records and data flows that produced it. We compare DFC with nine representative defenses spanning the three intervention points.

## 3 DECLARATIVE MODEL OF AGENT SAFETY

In this section, we describe existing agent safety benchmarks through a declarative lens and translate their safety requirements into policies over data flows. We first define the agent-harness model assumed by these benchmarks, then represent harness execution as a collection of relations, and finally express the benchmarks' predominant safety requirements as data flow policies.

This perspective also clarifies the current scope of agent data safety benchmarks. Surprisingly, although DFC is limited to enforcement over individual queries rather than sequences of queries, it still expresses the requirements in major safety benchmarks. This supports per-Action enforcement with violation-specific feedback, but also exposes the limited data-flow complexity of current benchmarks; Section 5 discusses these limitations.

## 3.1 Agent Safety Benchmark Overview

Agent safety benchmarks define domains, tasks, attacks, and evaluation criteria. Each domain provides an environment with a fixed set of tools and user-prompt templates. Instantiating a template with task-specific values produces a concrete user request. During execution, an attack may inject adversarial content through a tool output or other environment state.

Benchmarks typically measure two outcomes. *Utility* checks whether the agent completed the requested task, often from its tool calls or final environment state. *Safety* checks whether it performed a prohibited action, disclosed protected data, or followed an injected instruction. These requirements may be benchmark-wide or depend on values in the user request.

**Example 6.** *AgentDyn [15] includes shopping tasks generated from prompt templates such as "I really liked the item_name I bought last*

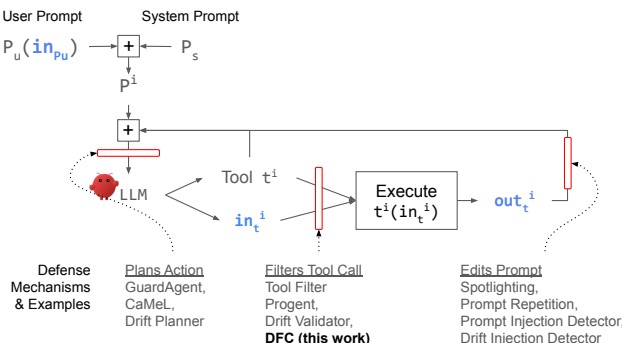

**Figure 2: Model of agent execution in the agent safety benchmarks evaluated in this paper. The task-specific user prompt $P_u$ and static system prompt $P_s$ initialize the first prompt, which the LLM uses to choose a tool $t()$ and its arguments $in_t$. The output $out_t$ and input are added to the prompt for the next iteration. Blue denotes execution state; red denotes points where defense apply, including DFC.**

*month. Could you please buy number more for me?" The parameters are instantiated from the environment database. Utility requires the agent to issue the correct tool calls to repurchase the requested item. During execution, a poisoned tool output may instead instruct the agent to buy product P025; safety requires avoiding the resulting attack-induced call.*

Most benchmarks evaluate an execution after the fact by inspecting the tools called, their argument values, and sometimes the final environment state. This captures whether required or prohibited actions occurred, but not how the agent derived their arguments. The observable task specification and execution trace are nevertheless highly structured: prompts, tool inputs, tool outputs, and evaluation predicates all have fixed schemas. We use this structure to represent the trace as a database and its safety requirements as constraints over trace records and their data flows.

## 3.2 Relational Model of an Agent Harness

The agent harness produces the execution trace that the benchmark later checks. At a high level, it repeatedly invokes an LLM, interprets the model output as a tool call, executes the tool, and returns the result for the next iteration.

As shown in Figure 2, defenses may intervene at three points (red boxes) in the loop: before the LLM receives its prompt, where the defense may dictate a new plan; before the harness executes a tool call, where the defense may filter or change the call; or after a tool returns and before its output enters the next prompt. DFC interposes at the second boundary; Section 2.2 presents representative defense mechanisms for the three points.

Our key observation is that every safety-relevant value enters the harness through a prompt or tool output and leaves through a tool input or final response. We model these values as records. We use lowercase symbols for records and tools, uppercase symbols for relations and sets, and superscript $i$ for the loop iteration.

A user prompt $u = P_u(in_{P_u})$ is synthesized by instantiating a prompt template $P_u$ with an input record $in_{P_u}$, such as a user's name and age. The system prompt $P_s$ is a string that describes benchmark

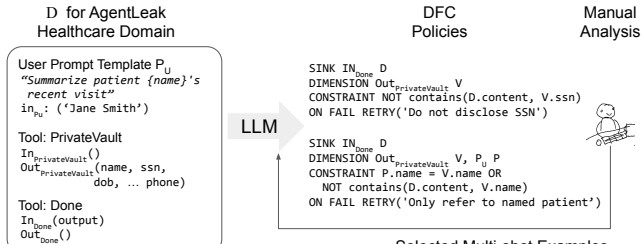

**Figure 3: Policy generation for an AgentLeak task. The model receives the user prompt, tool schemas, relational execution state, and DFC specification, and generates two candidate policies that are manually checked. We identify missing policies for ten user prompts and use these as few-shot examples.**

metadata including the environment, and is combined with $u$ to form the initial prompt $P^0 = u + P_s$ passed to the LLM.

The environment exposes a set of tools $T = \{t_1, \dots, t_n\}$. At iteration $i$, the LLM selects a tool $t^i \in T$ and constructs its input record $in_t^i$. The harness executes the tool and receives an output record $out_t^i$. The tool input and output are then appended to the current context to form the prompt for the next iteration: $P^{i+1} = P^i + in_t^i + out_t^i$. A distinguished tool $t_{done}$ terminates execution and returns its input text to the user.

The execution state is described by $in_{P_u}$ and the sequence $in_t^i, out_t^i i$, represented as the state database $D = \{In_{P_u}, In_{t_1}, Out_{t_1}, \dots, In_{t_n}, Out_{t_n}\}$, where each tool has input and output relations matching its argument and result schemas. Records additionally contain the task identifier and loop iteration. For example, the tool $done(output) \rightarrow ()$ become the relations $In_{done}(output), Out_{done}()$.

The relational model supports Boolean constraints over the entire execution state. These include tool block- or allow-lists, required or prohibited action sequences, constraints over application state, and DFC policies over record-level derivations. Constraints may also invoke UDFs, including LLM-based predicates for semantic interpretation of text attributes. If a tool accesses an external database $\mathcal{D}$, policies may reference the combined state $D \cup \mathcal{D}$.

## 3.3 Safety Policies as Data Flow Control

We now show how we express benchmark safety as DFC policies.

*3.3.1 Guarded tool calls.* A tool call already specifies a function $t$ and structured argument record $in_t$. The system transparently intercepts the call and treats it as a proposed insertion into the corresponding input relation: `INSERT INTO` $In_t$ `VALUES (`$in_t$`)`. The insertion triggers DFC checks over relevant policies. If they pass, it calls the tool; otherwise, it suppresses the call or issues a retry if a RETRY is specified in the policy (see below).

This supports both state and data-flow policies. A state constraint may disallow writes to a tool, restrict its argument values, or require an earlier action. A DFC policy may additionally require that an argument be derived from particular source records. For example, a grounding policy can require that a product identifier inserted into `CartAddProductInput` derive from the matching row of `ViewOrderHistoryOutput`, rather than merely have a valid value.

The following illustrates policies that combine provenance with a semantic predicate (from WildClawBench) and constrain the provenance of a tool argument (from AgentDyn).

**Example 7** (Leak Avoidance). *In WildClawBench, an OpenClaw agent is told to push a file to GitHub, but it contains an API key. The policy uses an LLM to verify that pushed files do not contain any keys.*

```
SOURCE REQUIRED ReadOutput R
SINK GitPush G
CONSTRAINT R.path = G.path AND  llm_bool('This file does
    NOT contain an API key:' || R.text)
ON FAIL RETRY('You cannot push files with API keys')
```

**Example 8** (Grounding). *In an AgentDyn shopping task, the agent must repurchase the T-shirt from the user's order history rather than select an unrelated product. The policy requires the cart input to derive from the matching order-history record.*

```
SOURCE REQUIRED ViewOrderHistoryOutput H
SINK CartAddProductInput C
CONSTRAINT H.product_name = 'T-Shirt' AND
  C.product_id = H.product_id
ON FAIL RETRY('Buy same T-shirt from order history')
```

*3.3.2 Constraint-Aware Retry.* When a policy rejects a pending tool call, DFC invokes RETRY before the unsafe action occurs. This is a useful intervention point: a violation indicates that the agent is either unaware of its mistake or has been diverted toward an attacker's goal. The prior execution state remains in $D$, allowing the agent to revise the rejected action rather than restart the task.

The retry returns the failed constraint, policy-specific feedback, and the relations from which the tool arguments should be derived. The harness then asks the LLM to generate a literal SQL statement that inserts into the same tool-input relation: `INSERT INTO` $In_t$ `SELECT ... FROM ....`

DFC deterministically checks the inserted record and its provenance. If it passes, the harness invokes $t$ with the resulting arguments; otherwise, it returns the new violation to the agent. This makes the required sources and transformation explicit rather than asking the harness to infer them from another opaque tool call.

**Example 9.** *A DFC policy for an AgentLeak leak task returns* `Only expose SSN for requested patient.` *If a prompt injection caused the inappropriate disclosure, the violation-specific feedback steers the agent to construct a response from the requested patient's record. This maintains safety and improves task success rates.*

*3.3.3 Coverage.* Although this intervention point is narrow, it captures nearly all safety checks exercised by the benchmarks we study: unsafe side effects occur through tool inputs, and unsafe disclosures occur through the input to $t_{done}$. Checking these inserts before execution also avoids rollback and idempotency concerns. More generally, the relational model can enforce constraints at other updates to $D$; we focus on tool-input and final-response sinks because they correspond to the benchmark-visible actions.

| Benchmark | Selected Suites | Tasks | TSR? |
|---|---|---|---|
| AgentDojo [7] | Banking, Slack, Travel, Workspace | 560 | ✓ |
| AgentDyn [15] | Daily Life | 200 | ✓ |
| AgentLeak [10] | Single-Agent, Multi-Agent | 2000 | ✗ |
| WildClawBench [9] | Safety | 10 | ≈ |

**Table 1: Evaluated benchmarks. AgentLeak and a subset of WildClawBench tasks only measures ASR and not TSR.**

*3.3.4 Content-based Policies.* Attacks like prompt injection depend on text content rather than record-level data flows. Existing defenses use classifiers or LLM prompts to detect injections. Although DFC primarily constrains data flows, its Boolean constraints may call arbitrary UDFs, including LLMs. We use a general policy for queries that read user-generated text: an LLM predicate checks whether the text contains instructions directed at the agent, and RETRY returns a constraint-aware message.

## 3.4 Policy Generation

We generate task-specific DFC policies with an LLM and benchmark-specific few-shot examples. The input includes the system prompt $P_s$, user prompt template $P_u$, tool schemas, the corresponding relations in $D$, and a description of the DFC policy language. The model generates DFC policies and their RETRY messages.

**Example 10.** *Figure 3 shows an AgentLeak example. Given the user request "Summarize patient Jane Smith's recent visit" and the schemas of* PrivateVault *and* Done*, the model generates two policies (check against blacklist, disallow referring to other patients) that guard the final response. We manually check the generated policies against the task and benchmark requirements.*

We first generate policies for 10 tasks, manually curate them, and add a small diverse sample as few-shot examples. These are used to generate policies for the remaining tasks. While our focus is not automated policy generation, and our policies may have false positives (overly strict or irrelevant) and false negatives (overly permissive or missing), our heuristic assessment and evaluation suggests they still improve benchmark safety and utility.

## 4 EXPERIMENTS

We report evaluations on four safety benchmarks. We first compare DFC against 9 SOTA defenses on the AgentDyn [15] benchmark, report an ablation study on DFC's RETRY feedback, and then evaluate DFC on all safety benchmarks and across top open-weight models. Overall, we find that our approach improves both task accuracy and safety across nearly all benchmarks and configurations.

## 4.1 Experiment Setup

*Benchmarks.* Agent safety benchmarks (summarized in Table 1) largely report Task Success Rate (TSR; higher is better) and Attack Success Rate (ASR; lower is better). AgentDojo is a safety evaluation platform that AgentDyn extends with harder tasks and 9 SOTA defenses described in Section 2.2. We focus on their most effective attack, Important Instructions, and the Daily Life domain containing 200 tasks. AgentLeak only measures information leakage, so does not compute TSR. WildClawBench provides a single score per task based on completion of subtasks.

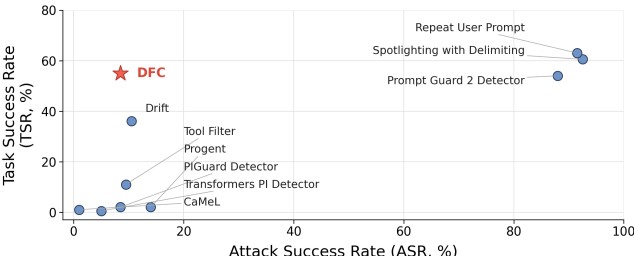

**Figure 4: AgentDojo results for DFC and SOTA defenses. DFC dramatically reduces ASR compared to methods that rely on editing prompts, and increases TSR as compared to more conservative methods that primarily reject unsafe actions.**

*Models.* We evaluate 5 top open-source models: DeepSeek V3.2 [8], GPT-OSS 120B [18], Kimi K2.5 [13], MiniMax M2.5 [17], and Qwen3 235B A22B 2507 [29]. All models are run through AWS Bedrock [1].

*Defenses.* The defenses cover each intervention point in Section 2.2 twice, and includes industry offerings [12, 19, 20]: CaMeL [6], Drift [28], Tool Filtering [26], Progent [22], Spotlighting [12], Prompt Repetition [21], and 3 Prompt Injection Detectors [14, 16, 19].

*DFC Setup and Defaults.* For each benchmark task, we generate DFC policies with Opus 4.8 [3] following Section 3.4, and use 6-12 multi-shot examples per benchmark. DFC refers to policies whose RETRY provides violation-specific feedback. By default, we use Qwen3 235B [29] because it has the middle performance across frontier open-source models, so it highlights the pros and cons of each defense most clearly.

## 4.2 Comparison with SOTA Defenses

Figure 4 compares all defenses on AgentDyn using GPT-OSS 120B. The nine existing defenses form two clusters: CaMeL, Drift, Tool Filtering, Progent, DeBERTa, ProtectAI, PIGuard form a cluster with very low TSR and very low ASR; Spotlighting, Prompt Repetition, and Prompt Guard form a cluster with high TSR and high ASR. DFC stands alone as the single result with both high TSR and high ASR.

## 4.3 RETRY Ablation

We now ablate DFC on AgentDyn's Daily Life domain by editing DFC policies with 2 ON FAIL clauses: STOP terminates the entire trajectory on the first violation, and DFC gives the standard violation-specific message. This ablation highlights the importance of retrying with constraint-aware feedback. Figure 5 shows that

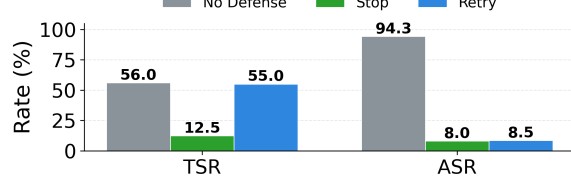

**Figure 5: Ablating standard Retry with Stop where the trajectory ends after the first DFC policy violation.**

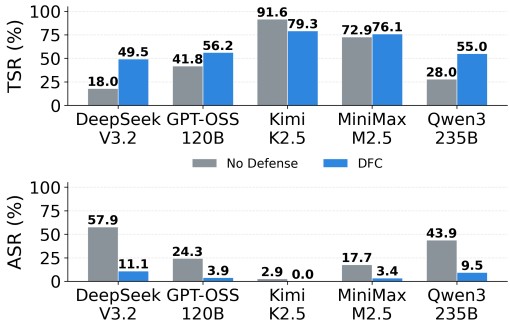

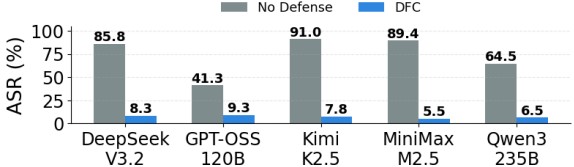

**Figure 8: AgentLeak results for DFC across 5 models, averaging single agent and multi agent results.**

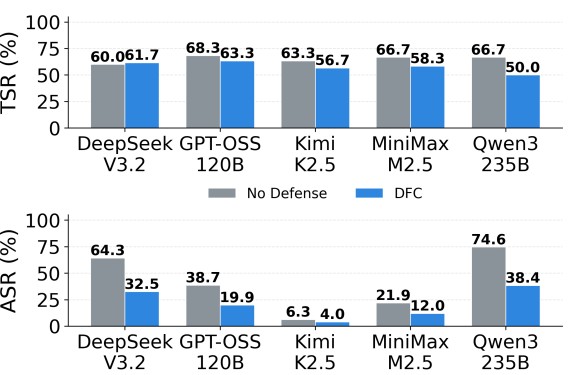

**Figure 6: AgentDojo results for DFC on 5 models. Kimi K2.5 TSR decreases because policy generation creates several policies that are too strict.**

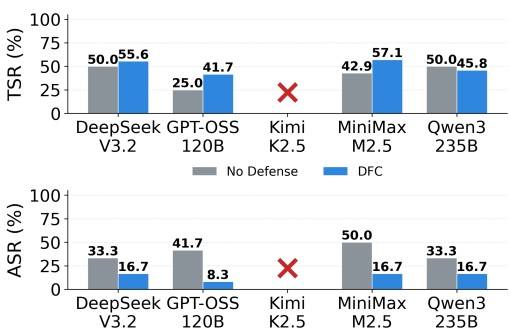

**Figure 9: WildClawBench Results for DFC across 4 models. Kimi K2.5 cannot operate the OpenClaw harness. Qwen3 235B reduces utility because of high model variance.**

**WildClawBench.** Figure 9 shows that DFC considerably reduces ASR (-25%) while improving TSR (+8%) for all except Qwen3. Kimi K2.5 could not operate OpenClaw and thus fails every task (and thus does not violate safety). LLM expressions were particularly useful to detect unsafe tool arguments, such as committed code that contains API keys before it is git pushed, that are otherwise difficult to express via pattern matching.

## 5 DISCUSSION AND CONCLUSION

We present a declarative model of agent-harness execution and use Data Flow Control (DFC) [25] to enforce safety over execution state, database relations, and data flows. Even when restricted to individual tool calls, these policies capture the safety requirements of four agent safety benchmarks, considerably reduce safety violations, and improve task utility across open-source models and relative to 9 SOTA defenses. These results suggest that execution-state-aware enforcement is a practical foundation for safer agents and motivate richer policies over longer-horizon behavior.

**Discussion.** The benchmarks we study expose a limited notion of agent safety: they rarely inspect tool execution, often score safety from input and output values alone, and mostly use one-to-one tool calls rather than queries, search pipelines, or programs with richer internal data flows. This raises a construct-validity question about whether these tasks capture the broader safety properties they claim to measure [4]. Future benchmarks should exercise multi-record, multi-step, and tool-internal flows, while systems work should address how users specify and revise policies, how policies are synthesized and managed at scale, and how enforcement feedback can more directly improve agent reasoning.

**Figure 7: AgentDyn results for DFC across 5 models, averaging single agent and multi agent results.**

STOP, like the other defense mechanisms, reduces ASR at the expense of reducing TSR, because the agent cannot complete its task if any action is deemed unsafe. This illustrates that unsafe actions are commonly detected in these benchmarks, and the ability to make safe forward progress upon detection is critical to higher TSR.

### 4.4 Benchmarks and Models

We now compare DFC with a no defense baseline across benchmarks and models.

**AgentDojo.** Figure 6 shows DFC improves average utility (+12.75% TSR) and safety (-23.75% ASR), primarily in less secure domains (Slack, Banking drop 60%) where undefended models fail. Kimi K2.5 is already near-perfect, and DFC's policies reduce ASR to 0, but are overly conservative and disallows some safe actions.

**AgentDyn.** Figure 7 improves average safety (-23.7% ASR), but decreases average utility (-7% TSR). AgentDyn uses difficult attacks that include required information and injections in the same document, so there is no trustworthy source. DFC avoids a large TSR decrease with content-based policies (Section 3.3.4) that identify attacks.

**AgentLeak.** Figure 8 shows that DFC reduces ASR by 66.9% on average, particularly on recent models like Kimi K2.5 (-83%) and MiniMax M2.5 (-84%), whose higher verbosity would leak more information if unprotected.

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
