# OpenReview forum: "Deterministic Data Flow Control Improves Agent Utility and Reduces Safety Violations"
_VLDB.org/2026/Workshop/NOVAS — NOVAS 2026_

### Official Review · Reviewer_VzpH · 2026-07-06

**Confidence:** 3

**Improvement Opportunities:**

- Translation of agent execution into relations is valid, but it feels a little contrived to be using literal SQL statements to enforce provenance (e.g., insert on retry). Non-database audience may find this quite unpalatable.
-  The evaluation section does not include measurements of overhead in latency, throughput, etc.
- Correctness of generated policies is a major vulnerability -- deterministic checking on weak or incorrect policies is not meaningfully stronger than LLM-based checks etc. Evaluation on the correctness of generated policies can be improved with experiments and more scenarios, rather that a simple "we check the output by hand"

**Minor Comments:**

- Figure 4 caption says AgentDojo while the text talks about AgentDyn
- Page 2 ofetn -> often

**Short Summary:**

The paper applies Data Flow Control from database prior work to agent. By translating agent execution, tool calls, and external exposure of results as database operations, this work enables deterministic checking of declarative constraints over execution state and provenance. The author additionally propose a retry-based mechanism that helps agents correct course with feedback rather than simply disallowing execution. The results show that this approach can reduce attack success rate without hurting task success rate on a variety of benchmarks and models.

**Strong Points:**

- The core observation that agentic safety requires provenance is valid and actionable
- Good results compared to baseline

---

### Official Review · Reviewer_QDzk · 2026-07-10

**Confidence:** 3

**Improvement Opportunities:**

O1: The paper is not self-contained, as the explanations of Data Flow Control are too short to understand. Understanding DFC is required to understand the paper. To understand DFC, the authors refer to a 13 page arxiv paper from the same author uploaded at the same time as this paper. To improve the paper, the authors should expand their explanation of DFC.

**Minor Comments:**

- A concrete example of how tools are translated to relations and how they are filled with concrete tuples with concrete values during agent execution is missing. The example where the tool done(output)-->() becomes relations In_done(output), Out_done() is too abstract. Instead, use a more useful tool for explaining, like DB querying, and show with a concrete example (e.g. querying SSN of patients), how the tool translates to relations, what their attributes are and how they are filled with values.

- Figure 2 shows that DFC is executed before Tool call, but in Example 5 the constraint is about the output of a tool call. My understanding is that the constraint in Example 5 is executed when the agent is finished (SINK IN_done D, where done is a pseudo-tool), where it has access to the output of the tool call. This is confusing. The authors should better explain when the constraints are checked.

**Short Summary:**

The authors propose Data Flow Control as an agent safety mechanism that allows users to declaratively express data flow constraints for agents. For instance, users can declaratively express that an agent should never reveal the SSN of other patients. For that, tool inputs and outputs are modeled as relations and DFC expresses constrains how input records can contribute to output records. DFC also has a retry directive that gives feedback to an agent when a DFC constraint is triggered, which increases the likelyhood that the agent can still solve the task successfully. The authors show that DFC is better in trading off task success rate and attack success rate than baselines.

**Strong Points:**

- Extensive Evaluation
- Many examples that help understanding
- The authors show that the proposed approach is best in trading of Task Success Rate and Attack Success Rate compared to several baselines.
- Agent safety is an important topic.
- The idea is highly interesting

---

### Official Review · Reviewer_Dorn · 2026-07-13

**Confidence:** 4

**Improvement Opportunities:**

1. In Section 3.4, the task-specific policies are generated using benchmark-specific examples and then manually checked against the benchmark requirements. This may encode some of the expected behavior into the defense. It would help to evaluate policy generation on held-out tasks and to say more clearly which policies or examples were manually edited.

2. Some claims about determinism need to be clarified. Section 2.1 says that constraints may invoke "LLM-based semantic predicates" while enforcement remains deterministic, and Section 3.3.4 uses an LLM predicate to identify instructions in user-generated text. The enforcement procedure may be deterministic given the predicate's output, but the complete safety decision is not. The paper should distinguish purely relational policies from policies that depend on an LLM predicate.

3. This work reads primarily as an application of DFC to agent safety. Prior work [2,24,25] has already established much of the DFC policy model and motivated agent-facing enforcement. I would like the paper to be more explicit about what is new here.

**Minor Comments:**

- Table 1 mixes user tasks, security test cases, scenarios, and execution paths. Please label these units more precisely and explain how TSR and ASR are computed for each benchmark.
- Table 1 reports 560 AgentDojo tasks, but AgentDojo contains 97 user tasks and 629 security test cases according to their paper. Am I mistaken?
- Section 4.2 says DFC achieves "both high TSR and high ASR"; you likely meant high TSR and low ASR.
- Section 4.2 identifies Figure 4 as AgentDyn, while its caption says AgentDojo.

**Short Summary:**

This paper applies Data Flow Control (DFC) to agent safety. By modeling agent-harness execution as relational state, DFC can enforce constraints over actions, execution history, and the data used to produce tool arguments. The paper also introduces RETRY, which gives agents constraint-specific feedback after blocking an unsafe action. I like the paper, and it addresses an important problem.

**Strong Points:**

1. The evaluation is thorough and has clear wins. It covers multiple benchmarks, models, and defenses, and the results show substantial safety improvements.

2. Agent safety is a clear and important problem, especially as agents invoke tools and modify external state.

3. The system design and methods are sound.

4. Well written and structured.

---

### Decision · Program_Chairs · 2026-07-16

**Decision:**

Accept

**Comment:**

This paper applies Data Flow Control to agent safety through declarative constraints over tool actions, execution history, and data provenance. The extensive evaluation across multiple benchmarks and models shows strong reductions in attack success while preserving task success, and the RETRY mechanism provides a practical way for agents to recover from blocked actions. We look forward to valuable discussions on agent safety at the workshop.